# Examining Emotional and Physical Burden in Informal Saudi Caregivers: Links to Quality of Life and Social Support

**DOI:** 10.3390/healthcare12181851

**Published:** 2024-09-14

**Authors:** Wejdan Homid Aloudiny, Ftoon Fahad Alsaran, Fawziah Muqbil Alessa, Fatmah Almoayad, Lamiaa Fiala

**Affiliations:** Department of Health Sciences, College of Health and Rehabilitation Sciences, Princess Nourah Bint Abdulrahman University, P.O. Box 84428, Riyadh 11671, Saudi Arabia; 442005206@pnu.edu.sa (F.F.A.); 442002171@pnu.edu.sa (F.M.A.); faalmoayad@pnu.edu.sa (F.A.); lefiala@pnu.edu.sa (L.F.)

**Keywords:** caregiver, informal caregiver, caregiver burden, social support, quality of life

## Abstract

Background: Caregiver burden is an important issue for individuals who tend to be chronically ill, disabled or elderly family members. This burden affects caregivers around the world and can have a negative impact on their physical and mental health, ultimately reducing their quality of life. Methods: This study of informal adult caregivers in Saudi Arabia used a cross-sectional analytic design to explore the connections between caregiver burden, social support and quality of life. Data were collected using nonprobability convenience sampling through electronic questionnaires in Arabic. The Zarit Caregiver Burden Interview (ZBI-12), the Oslo Social Support Scale (OSSS-3) and the SF-12 Health Survey (short form of 12 questions) were used to assess caregiver burden, social support and quality of life, respectively. The relationships between these variables were analysed, and the statistical significance of the findings was reported. Results: The study revealed a connection between caregiver burden and both social support and quality of life. Caregivers with moderate to high burdens often had poor social support (60.52%) and a lower quality of life (72.47%). A statistically significant negative correlation between caregiver burden and quality of life (*p* < 0.05) indicated that caregivers with a higher burden had a lower quality of life. Similarly, a statistically significant negative correlation was found between social support and caregiver burden (*p* < 0.05), suggesting that caregivers with more social support experienced a lower burden. Conclusions: A higher caregiver burden is linked to a lower quality of life, especially when social support is inadequate. These findings highlight the need for targeted interventions to reduce caregiver burden by enhancing social support systems for caregivers and improving their quality of life. Recommendations include the development of community-based support programmes and policy changes to support informal caregivers.

## 1. Introduction

“Caregiver burden can be defined as the strain or load borne by a person who cares for a chronically ill, disabled or elderly family member” [1] (p. 429). The burden of caregiving is an established global phenomenon [2]. According to some literature, family caregivers frequently endure psychological, behavioural and physiological impacts, such as depression, anxiety, high levels of negativity, impulsive temperament, fatigue, sleep issues, increased risk of illness and injury, mortality, a weakened immune system, and increased usage of medical services and medications [3,4]. While caregiver burden is a global issue, the specific cultural and social dynamics within Saudi Arabia, such as traditional family structures and societal expectations, can uniquely influence the caregiving experience. However, there is limited research that addresses these factors in the Saudi context.

Studies have found that caregivers for persons with Alzheimer’s disease, dementia with Lewy bodies (LBD) and other chronic conditions often experience moderate to high caregiver burden, which has been associated with an elevated risk of psychiatric problems, neuropsychiatric symptoms and reduced functioning in daily activities [5,6].

Caregivers of individuals with disabilities face a range of physical, emotional, financial and social difficulties. Research has demonstrated that factors associated with caregiver burden include the education and family income of caregivers, as well as characteristics of care recipients, such as age, education, employment and types of sensory or visual impairments; both types of factors play significant roles in the level of burden experienced [7].

A multi-centre cross-sectional survey conducted in four hospitals in China revealed a significant negative correlation between caregiver burden and quality of life among caregivers; as the caregiver burden increased, the reported quality of life among caregivers decreased [8]. Similar findings were observed in a study conducted in India, which highlighted that a higher perceived caregiver burden was associated with impaired quality of life across psychological, social and environmental domains [9].

In Saudi Arabia, studies have shown that the caregiver burden is prevalent among those caring for patients undergoing haemodialysis and stroke survivors, leading to considerable physical and psychological stress [10,11]. However, there is a lack of comprehensive studies that have explored how caregiver burden interacts with social support and quality of life, particularly in the context of chronic illnesses within Saudi society.

Coping strategies are essential for managing challenges perceived as overwhelming or exceeding an individual’s resources. In stressful situations, coping methods aim to solve the problem or control the emotional response to it [12]. A particularly valuable type of coping strategy is the use of social support. Social support has been described as “support accessible to an individual through social ties to other individuals, groups, and the larger community” [13] (p. 194). The literature indicates that social support and preparedness are protective factors in reducing caregiver burden and improving caregivers’ quality of life. It has been suggested that the level of social support is an important indicator of caregivers’ quality of life and that higher levels of social support help caregivers prepare themselves for their new responsibilities. Despite the global recognition of these factors, their specific impact on informal caregivers in Saudi Arabia remains underexplored, particularly concerning how traditional social networks and cultural expectations shape the caregiving experience. In recent years, web-based interventions using videos and informational texts have also been offered to improve caregiving readiness and provide social support [8,14].

Additional research shows that social support is crucial for improving psychosocial health and promoting positive adaptation among caregivers. Adequate social support serves as a protective factor for mental health, directly affecting the quality of life and psychological state of both caregivers and patients [15,16]. However, the specific dynamics of social support in the Saudi context, where family ties and community networks are integral, have not been sufficiently studied.

Informal caregivers, who are often family members or friends, face a range of responsibilities that can be overwhelming. These duties can include personal care tasks, such as bathing, dressing and feeding an individual; managing medications; providing emotional support; and coordinating with healthcare providers [17,18]. In Saudi Arabia, where family members are typically expected to take on caregiving roles, these responsibilities can be particularly taxing, especially for those who may also be balancing work and other familial obligations [19]. The physical and emotional demands of caregiving, combined with limited external support, can lead to considerable strain and contribute to caregiver burden [20].

Caregiving also has financial implications. Many caregivers incur out-of-pocket expenses for medical supplies, transportation and other caregiving-related costs. In some cases, caregivers may need to reduce their work hours or leave their jobs entirely, leading to lost income and increased financial stress [21]. In Saudi Arabia, where caregiving is often considered a familial duty rather than a paid occupation, the financial burden on caregivers can be substantial, particularly in households with limited incomes.

Quality of life (QoL) is defined as an individual’s perception of their position in life, influenced by their culture, value systems, goals, expectations, standards and concerns [22]. QoL encompasses physical, psychological, social, environmental and spiritual domains. The literature demonstrates that the caregiver burden greatly impacts the QoL of caregivers and includes factors such as the caregiver’s health, financial situation and social support, along with the degree of dependency of an individual on the caregiver, all playing a role [23]. In Saudi Arabia, understanding these dynamics is essential for developing effective interventions and support systems tailored to the specific needs of caregivers within this cultural context.

Accordingly, the current authors assumed that social support would directly influence both caregiver burden and quality of life. Specifically, higher levels of social support were expected to reduce caregiver burden and, in turn, enhance the quality of life among informal caregivers (Figure 1). The interrelationships among these factors suggest that improving social support networks could mitigate the negative impacts of caregiving stress, thereby improving overall well-being.

The aim of this study was to assess caregiver burden and examine its relationships with quality of life and social support among informal caregivers in Saudi Arabia. This study sought to fill a gap in the current literature by providing insights into the unique challenges faced by Saudi caregivers, particularly within the context of chronic illnesses. The study tested the assumption of a relationship between low caregiver burden and high quality of life and social support. To achieve this, the study addressed the following research questions:What are the relationships between caregiver burden, social support and quality of life among informal caregivers in Saudi Arabia?Are there significant differences in caregiver burden, social support and quality of life based on demographic factors such as gender, age, education and marital status among informal caregivers in Saudi Arabia?

## 2. Materials and Methods

A cross-sectional analytic study design was used to assess the level of caregiver burden and examine its relationship with quality of life and social support among informal Saudi caregivers in Saudi Arabia. The study was conducted from October 2023 to May 2024.

This study focused on adults of both genders residing in Saudi Arabia. Participants had to be informal caregivers, defined as unpaid individuals, such as family members, friends or neighbours, who cared for loved ones. Formal caregivers were excluded from the study.

The minimum sample size was calculated as part of a power analysis to ensure that the study had sufficient power to detect significant relationships among caregiver burden, social support and quality of life. The formula *n* = *Z*^2^*PQ*/*d*^2^ was used, where *n* = desired sample size, *Z* = standard normal deviation, *P* = proportion of the characteristic under study, *Q* = 1 − *P* (proportion in the target population not having the particular characteristic) and *d* = degree of accuracy. Assuming an estimated proportion of 50% due to the lack of studies, a confidence level of 1.96 and a degree of accuracy of 0.05, the calculated desired sample size was 385 participants. To account for potential non-response and dropout rates, a 10% adjustment was made, increasing the final sample size to approximately 424 participants to ensure sufficient power. The time required to complete all the questionnaires was approximately 7–15 min. The participants were able to complete all the questionnaires in a single session, ensuring consistency in their responses.

A nonprobabilistic convenience sampling technique was employed. The questionnaire was constructed using Google Forms, and participants were recruited through social media platforms, such as Twitter, WhatsApp and Telegram, where the research team disseminated the questionnaire widely. The data collection process involved clear instructions provided at the beginning of the survey to guide participants through each section. Anonymity and confidentiality were maintained to encourage honest responses. Reminders were sent to potential participants to increase the response rate, and all survey questions were marked as mandatory, leaving no missing data in submitted responses.

While convenience sampling may introduce potential biases and limit generalisability, efforts were made to mitigate this by ensuring broad distribution across various social media platforms and demographic groups. To further reduce potential response bias, neutral language was used to avoid leading questions, and balanced response options were offered on Likert scales and other multiple-choice questions to fairly represent a selection of viewpoints. A pilot test was conducted with a small sample to identify and address any potential biases or confusing questions, and the questionnaire was refined based on feedback from this testing.

Three valid scales were used in the questionnaire to collect the data. They covered the three major constructs: caregiver burden, social support and quality of life. Sociodemographic data were also collected: age, gender, the caregiver’s relationship with the patient, educational level, monthly income, marital status, type and severity of disease, number of people requiring care, receiving support in caregiving tasks, frequency of care and average time spent on caregiving tasks. The three scales were the Zarit Caregiver Burden Interview (ZBI-12), the Oslo Social Support Scale (OSSS-3) and the SF-12 Health Survey (short form of 12 questions, as a measure of quality of life).

### 2.1. Zarit Caregiver Burden Interview (ZBI-12)

This was a validated Arabic questionnaire. The short version of the ZBI comprises 12 questions encompassing the personal strain and role strain domains. Example questions include “Do you feel you do not have enough time for yourself?”, “Do you feel stressed between caring for and meeting other responsibilities?” and “Do you feel angry when you are around your relatives?”. Each question was scored on a five-point Likert scale ranging from 0 (never) to 4 (almost always). The cumulative score could range from 0 to 48. A higher score indicated a greater sense of burden. The total score was then placed into one of two categories of caregiver burden: 0–9 indicated no to mild burden, and 10 and above indicated moderate to high burden. This tool was previously tested for reliability and achieved a Cronbach’s alpha score of 0.77 [24]. 

### 2.2. Oslo Social Support Scale (OSSS-3)

This questionnaire was translated into Arabic and then reviewed and modified by two experts. The scale assesses the level of social support that caregivers have. It consists of three questions: “How many people are so close to you that you can count on them if you have great personal problems?”, “How much interest and concern do people show in what you do?” and “How easy is it to get practical help from neighbours if you should need it?”. The first question had four possible responses, the second question had five responses, and the third question had five responses. The responses were scored from one point to the maximum for the question (e.g., a question with four possible responses was worth 1 to 4 points, whereas a question with five possible responses was worth 1 to 5 points), with a total score of the scale ranging from 3 to 14 points. Higher scores represented high levels of social support, and lower scores represented low levels of social support. The total score was placed into one of three categories of social support: 3–8 indicated poor social support, 9–11 indicated moderate social support, and 12–14 indicated strong social support [25].

### 2.3. SF-12 Health Survey (Short Form of 12 Questions)

The validated Arabic version of the SF-12 Health Survey (hereafter SF-12) was used to assess the quality of life of the participants (with a Cronbach’s alpha value of 0.84, indicating very good reliability). The 12 items in the SF-12 measure eight subscales: physical functioning (PF), role limitations due to physical health problems (RP), bodily pain (BP), general health perceptions (GH), vitality (VT), social functioning (SF), role limitations due to emotional problems (RE) and mental health (MH). Four questions had two responses, two questions had three responses, three questions had five responses and three questions had six responses. The responses were scored from one point to the maximum for the question (e.g., a question with two possible responses was worth 1 or 2 points, whereas a question with six possible responses was worth 1 to 6 points), with total responses ranging from 12 to 47 points. The scores were categorised using quartiles to assess the quality of life. Scores in the top two quartiles (50% and above) indicated a higher quality of life, while those in the bottom two quartiles (below 50%) indicated a lower quality of life [26].

The three selected tools were combined into a single questionnaire and translated into Arabic using a forward–backward translation process to ensure accuracy and cultural relevance in the language used. The questionnaire was then reviewed by an expert panel consisting of three public health experts whose feedback confirmed the appropriateness of the tools. The translated questionnaire was then piloted with 10% of the sample, which allowed us to identify and refine any language that lacked clarity or could potentially confuse the respondents. The changes made following the pilot were minor, allowing the pilot sample data to be included in the analysis of the final study. Additionally, the reliability of the questionnaire was assessed using Cronbach’s alpha reliability test, which resulted in an alpha score of 0.59, indicating moderate internal consistency.

Statistical analysis was performed using the JMP software version 17 [27]. Descriptive statistics were employed to summarise the data, including the creation of frequency tables and figures to visualise the distribution of the variables. Pearson correlation coefficients were calculated to determine the strength and direction of the relationships between burden and quality of life, as well as burden and social support. Chi-square tests were conducted to investigate the associations between the burden categories and both quality of life and social support. A multiple logistic regression model was employed to predict caregivers’ burden based on several predictor variables. The dependent variable was the binary classification of caregiver burden, coded as 0 for no to mild burden and 1 for moderate to severe burden. The predictor variables, or independent variables, included age, gender, marital status, education, relationship to the patient, frequency of performing caregiving tasks, type of patient’s illness (acute or chronic), severity of the patient’s illness, number of people being cared for, level of social support and the time needed to perform care. To determine the most significant predictors, a backward stepwise (conditional) method was used for the variable selection. The final model included the following variables: gender, severity of illness, number of people being cared for, time needed to perform care and level of social support.

## 3. Results

The survey was completed by 426 participants; however, 23 were excluded for not meeting the study criteria (e.g., they were residents outside Saudi Arabia, were formal caregivers or did not sign the research consent form). Thus, the total number of participants included in the study was 403.

Table 1 provides a detailed breakdown of the caregivers’ burden scores according to the participants’ sociodemographic characteristics. All age groups reported experiencing a moderate to high burden, but a larger proportion of middle-aged to older individuals (60.53%) were affected at these levels. A higher percentage of female caregivers (58.1%) reported a moderate to high burden than males. A majority of caregivers with less education (58.8%) reported experiencing no to mild burden, whereas those with more education tended to report suffering from a moderate to high burden. All the caregiver groups reported experiencing a moderate to high burden, but a higher proportion was reported among the married (59.1%) and divorced/widowed (64.4%) participants and those with higher income (65.7%).

Table 2 provides the statistical breakdown of the caregiving duties reported by the participants. Most of the study’s participants reported taking care of individuals with chronic illnesses (81.4%). The greatest number of participants reported the severity of the patient’s illness to be moderate (46.9%), with almost as many reporting severe (32.3%). For the most part, the number of people requiring the attention of the caregivers was a single individual (69.9%). The majority of the participants reported receiving support for caregiving duties (81.01%). Daily care was the most prevalent frequency of caregiving (67.0%). The average time spent on caregiving tasks was reported as 1 h (38.9%).

In terms of caregiver burden classifications, the results showed that more than half of the caregivers (54.6%) experienced moderate to high burdens, while 46.4% experienced mild burdens. The results also revealed that most of the caregivers reported poor social support (57.8%), while 38.5% had moderate social support and only 3.7% had sufficient support. Regarding the level of quality of life among caregivers, the largest group reported a higher quality of life (55.8%), while 44.2% reported a lower quality of life.

To partially address the first research question regarding the relationships among caregiver burden, social support and quality of life, chi-square tests were performed on the categorised values of caregiver burden with the similarly characterised values of social support and quality of life (Table 3). The tests demonstrated significant associations between caregiver burden and both social support (*p* = 0.0171) and quality of life (*p* < 0.0001). Caregivers with no to mild burden predominantly reported moderate social support (54.2%) and higher quality of life (59.6%). In contrast, caregivers with moderate to high burdens mostly reported poor social support (60.5%) and lower quality of life (72.5%).

Table 4 further supports these findings by showing a statistically significant moderate negative correlation between caregiver burden and quality of life (r = −0.42, *p* < 0.05), suggesting that quality of life decreases as burden increases. In addition, it shows a significant weak negative correlation between social support and caregiver burden (r = −0.18, *p* < 0.05), suggesting that as social support increases, the burden on caregivers decreases.

In addressing the second research question, regarding significant differences in caregiver burden, social support and quality of life based on demographic factors, various results revealed several noteworthy patterns. Table 1 shows that female caregivers and those with higher education levels reported experiencing greater burdens, while male caregivers and those with lower education levels reported lower levels of social support. These differences were further explored using logistic regression to provide a more in-depth analysis (Table 5). The model revealed that several factors significantly predicted caregiver burden, including gender, the severity of the illness, the number of individuals under the caregiver’s responsibility and the amount of time dedicated to caregiving tasks.

## 4. Discussion

This study aimed to examine the relationship between caregiver burden, quality of life and social support among informal caregivers in Saudi Arabia, particularly within the context of chronic illnesses. Specifically, the study tested the assumption that higher levels of social support would reduce caregiver burden and, in turn, enhance quality of life. The study addressed two main research questions: (1) the relationship among caregiver burden, social support and quality of life and (2) whether demographic factors, such as gender, age, education and marital status, influenced these variables.

The results from the 403 participants revealed several key findings. In line with the first research question, the study found a significant moderate negative correlation between caregiver burden and quality of life, supporting the assumption that increased caregiver burden reduces quality of life. This finding is consistent with research conducted in Turkey on caregivers of stroke patients, underscoring the psychological and physical toll of caregiving, which is exacerbated by inadequate support systems, reinforcing the notion that as caregiver burden increases, quality of life decreases [28].

The results of the current study showed that the majority of study caregivers reported poor social support. This is consistent with a 2021 study in Turkey in which decreased perceived social support was linked to higher caregiver burden [29]. However, the weak negative correlation between caregiver burden and social support in this study could suggest that the relationship between these two variables may not be linear. It is possible that social support plays a mediating role in the relationship between quality of life and caregiver burden, warranting further investigation into these potential dynamics.

This relationship likely reflects the psychological and physical strain of caregiving, which is exacerbated by inadequate support systems.

The weak negative correlation between caregiver burden and social support suggests that, while social support does help alleviate some of the burden, its impact might not be as strong as expected. This resonates with previous literature [29] and may be due to the quality rather than the quantity of social support, highlighting the need for more personalised and culturally appropriate interventions. The significant relationships of caregiver burden with social support and quality of life highlight the compounded effect of these factors, consistent with research showing that perceived social support can mitigate caregiver burden through emotional support, practice assistance, stress relief, shared experience and help in self-care, thus enhancing caregiver quality of life [30,31].

The study also explored the second research question regarding demographic factors. In the current study, females reported higher caregiver burden levels than males. This is consistent with a 2022 study conducted in Lebanon that showed that female caregivers experience a greater burden than male caregivers [32]. This finding is particularly significant given that only 28% of the respondents in this study were male. In the Saudi context, this may reflect cultural norms and expectations around caregiving roles, which often fall more heavily on women. In addition, caregiver burden was found to increase with the severity of the patient’s illness. This was consistent with a 2016 study conducted in Egypt in which a direct relationship was found between caregiver burden and the physical and mental health status of the patients [33]. Moreover, the current study showed that the time needed for caregiving tasks significantly influenced caregiver burden. This is consistent with a 2021 finding from the United States in which caregiver burden was found to be related to the length of time involved in providing care; in that study, when caregiving hours reached 14 h per day, burden levels were reported to be the highest [34]. These findings emphasise the need for targeted interventions that consider the specific needs and circumstances of different caregiver groups.

Additionally, 75% of the respondents were single, which may indicate a cultural trend in the Middle East, where single individuals are more likely to remain in the family home and be available to provide care for family members in need. Furthermore, the high percentage (75%) of university graduates among respondents suggests that lower-educated individuals responded much less to the survey, possibly due to lower engagement with social media or other factors. Future research should explore these trends further to understand which groups of informal caregivers face the greatest burden and how policies can be tailored to meet their needs.

The main limitation of this research arose from the inability to obtain a specific population frame due to the lack of formal data and specific settings for informal caregivers in Saudi Arabia. As a result, a nonprobability convenience sample was used for data collection, which can limit the sample representation and generalisability of the findings. This sampling method, while practical, can introduce potential biases and restrict the extent to which the results can be applied to the broader population of Saudi informal caregivers. To mitigate these potential limitations, efforts were made to ensure a broad distribution across various social media platforms and demographic groups, as well as careful validation of the data collection and measurement through pilot testing, expert review and reliability testing. Despite these efforts, future studies should aim to use a probability sampling technique, if feasible, to enhance the representativeness of the data and generalisability of the findings. Qualitative research could also provide deeper insights into caregivers’ personal experiences, emotions and challenges, capturing the complexity of caregiver burden beyond numerical data.

## 5. Conclusions

Overall, this study highlights the significant burden faced by informal caregivers in Saudi Arabia, particularly among female caregivers and those with limited social support. The findings supported the assumptions based on the literature by demonstrating a strong link between higher levels of social support and reduced caregiver burden, which, in turn, is associated with a higher quality of life. The findings highlight a strong link between caregiver burden and lower quality of life, emphasising the critical role of social support in alleviating this burden. These results suggest the need for targeted interventions that enhance social support systems and consider cultural dynamics in caregiving roles.

To address these challenges, it is recommended that future efforts focus on developing culturally appropriate strategies to support caregivers. This could include creating community-based programmes that offer practical assistance and emotional support, increasing public awareness about the importance of caregiving, and implementing policies that provide financial and institutional support for informal caregivers. These steps could ultimately improve caregivers’ quality of life and reduce the burden they experience, contributing to better outcomes for both caregivers and those individuals for whom they provide care. Follow-up research should also consider both quantitative and qualitative approaches to provide a more comprehensive understanding of caregiver burden and support systems in Saudi Arabia.

## Figures and Tables

**Figure 1 healthcare-12-01851-f001:**
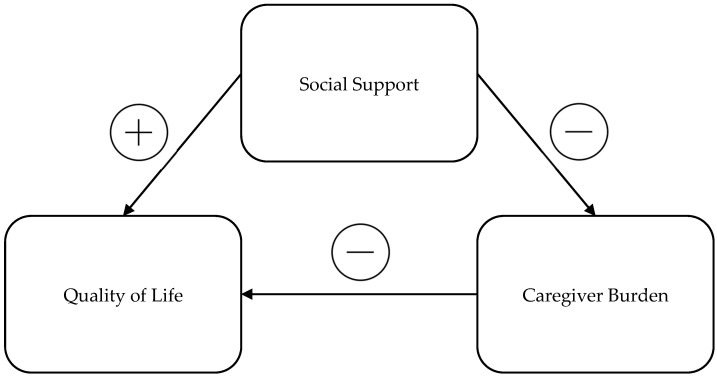
Social support impact framework.

**Table 1 healthcare-12-01851-t001:** Caregiver burden by sociodemographic characteristics (N = 403).

Characteristics	Caregiver Burden [N (%)]		Total (%)
	No/Mild	Moderate/High	
Age			
Young adult (18–25)	126 (46)	148 (54)	274 (68.0)
Adult (26–44)	42 (46.2)	49 (53.8)	91 (52.6)
Middle to old age (45+)	15 (39.47)	23 (60.53)	38 (9.4)
Gender			
Female	122 (41.9)	169 (58.1)	291 (72.2)
Male	61 (54.5)	51 (45.5)	112 (27.8)
Educational level			
Less than high school	10 (58.8)	7 (41.2)	17 (4.2)
High school or diploma	41 (48.8)	43 (51.2)	84 (20.9)
University and higher	132 (43.7)	170 (56.3)	302 (74.9)
Marital status			
Single	142 (47.2)	159 (52.8)	301 (74.7)
Married	36 (40.9)	52 (59.1)	88 (21.8)
Divorced/Widowed	5 (35.7)	9 (64.3)	14 (3.5)
Monthly income (Saudi Riyal)			
<8000	141 (47.8)	154 (52.2)	295 (73.2)
8000–16,000	30 (41.1)	43 (58.9)	73 (18.1)
>16,000	12 (34.3)	23 (65.7)	35 (35.7)

**Table 2 healthcare-12-01851-t002:** Caregiving duties reported by the participants.

Caregiving Duties	N (%)
Type of patient illness (N = 403)	
Acute	75 (18.6)
Chronic	328 (81.4)
Severity of patient illness (N = 403)	
Severe	130 (32.3)
Moderate	189 (46.9)
Mild	84 (20.8)
Number of patients requiring care (N = 316)	
1	221 (69.9)
2	67 (21.2)
3+	28 (8.9)
Receiving support with caregiving duties (N = 316)	
Yes	256 (81.0)
No	60 (19.0)
Frequency of care provided (N = 316)	
Daily	211 (67.0)
2–4 times per week	58 (18.4)
Once a week	13 (4.1)
2 times or less per month	33 (10.5)
Average time spent on caregiving tasks (N = 316)	
1 h	123 (38.9)
2–3 h	112 (35.5)
4+ h	81 (25.6)

**Table 3 healthcare-12-01851-t003:** Categorizations of caregiver burden by levels of social support and quality of life with chi-square results (N = 403).

	Caregiver Burden				Chi-Square Test
	No to Mild Burden		Moderate to High Burden		
	N	%	N	%	
Caregiver’s social support					
Strong social support	7	46.7	8	53.3	X^2^ = 8.13
Moderate social support	84	54.2	71	45.8	*p* = 0.0171 *
Poor social support	92	39.5	141	60.5	
Caregivers’ quality of life					
Higher quality of life	134	59.6	91	40.4	X^2^ = 41.12
Lower quality of life	49	27.5	129	72.5	*p* < 0.0001 *

* Statistically significant (*p* < 0.05).

**Table 4 healthcare-12-01851-t004:** Pearson correlation coefficients of caregiver burden scores with quality of life and social support (N = 403).

Scales	Pearson Coefficient (r)	*p*-Values
Burden and Quality of Life	−0.4288	*p* < 0.0001 *
Burden and Social Support	−0.1808	*p* = 0.0003 *

* Statistically significant (*p* < 0.05).

**Table 5 healthcare-12-01851-t005:** Results of the logistic regression analysis using a backward stepwise (conditional) procedure for determining the best predictors of caregiver burden.

Variables in the Model	B	S.E.	Wald	df	Sig.	OR (Exp(B))	95% CI for OR (EXP(B))	
							Lower	Upper
Gender	0.610	0.264	5.329	1	0.021	1.841	1.097	3.090
The severity of illness: 1 Severe (reference)			11.678	4	0.020			
2 Moderate	1.230	0.453	7.373	1	0.007	3.421	1.408	8.312
3 Mild	0.539	0.568	0.901	1	0.343	1.714	0.563	5.213
How many people are you caring for:								
(1) (reference)			5.546	3	0.136			
(2)	−21.159	21,454.161	0.000	1	0.999	0.000	0.000	
(3+)	−20.643	21,454.161	0.000	1	0.999	0.000	0.000	
The time needed to perform care:								
Daily (reference)			12.001	2	0.002			
2–4 times per week	−0.710	0.318	4.983	1	0.026	0.492	0.264	0.917
Once a week or less	0.263	0.330	0.636	1	0.425	1.301	0.681	2.484
Level of social support Poor (reference)			5.170	2	0.075			
Moderate	0.671	0.782	0.736	1	0.391	1.956	0.422	9.057
Sufficient	1.165	0.773	2.271	1	0.132	3.205	0.705	14.578
Constant	19.598	21,454.161	0.000	1	0.999	324,474,103.755		

## Data Availability

Data can be obtained by contacting the corresponding author if desired.

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
