# Peer review of "Examining Emotional and Physical Burden in Informal Saudi Caregivers: Links to Quality of Life and Social Support"

_healthcare, 2024, doi:10.3390/healthcare12181851_

Round 1
Reviewer 1 Report
Comments and Suggestions for Authors
While it is a good endeavor to examine the relationships between informal care, care burden and quality of life, based on the current article it is unclear for what reasons this research was designed in Saudi Arabia. A clear problem statement is missing, by which I mean that it is not clear against the background of what developments in Saudi Arabian society the research took place. Have there been changes in cultural and social conditions in the past decades that have necessitated research into the conditions of informal care. Have forms of cohabitation changed, has society become more individualistic, have care views changed, have relations between generations shifted?
The current introduction cites numerous studies from various countries involving very different diseases, conditions, and stages of disease (chronic or otherwise). Figures on incidence and prevalence of chronic disease processes and informal caregiving in Saudi Arabia are lacking. It is also unclear what questions and problems arise among informal caregivers specifically in the Saudi Arabian context. It would therefore be highly desirable to appreciate the studies, almost all of which have been conducted in other cultural and social contexts, against the background of the Saudi Arabian context.
In fact, in a situation where hardly anything is known about the questions and problems of these informal caregivers, qualitative research should be the first to be done. Then, based on a purposive sample, it can become clear what those questions and problems are, and what factors/variables play a role in this. Against this background, and with the help of subsequent literature research, a hypothesis can then be formulated that can be tested in a quantitative study. In doing so, the various factors/variables could be placed into a hypothetical model that goes beyond the research question of whether care burden and social support affect the quality of life of informal caregivers. That question can be answered in the affirmative prior to the study - given the literature that has already demonstrated these correlations in a variety of places around the world.
The question, however, is how to further differentiate the variables "care burden" and "social support," and how to connect them to "quality of life" in a model. After all, what exactly is care burden, and what is social support? And for that matter, what is quality of life? What aspects of these three variables are related, and in what ways? Is there mediation and/or moderation? And how can the variables be categorized into a clear model, and in what way (i.e. with what instruments) can this model be tested?
In my opinion, the current research question is not precise enough (see lines 101-102). That the care burden in Saudi Arabia can be determined (line 103) seems impossible given the "convenience sample" which is not at all representative of the population of Saudi Arabia.
What is listed in the article under the heading "literature review" is apparently intended to clarify the concepts of "informal caregiver," "care burden," "social support," and "quality of life. However, this constitutes a major overlap with the first part of the introduction, and furthermore, the concepts are not clarified and established. On the basis of these concepts, it is precisely in the methodological part that their operationalization could take place, in line with the hypothetical model of the relationships, and of course the choice of instruments appropriate to the concepts and their operationalization.
As mentioned, I miss the connection of the basic concepts to the instruments, at least the connection of the different aspects of care burden, of social support and of quality of life. For example, what aspects of care burden are really burdensome? Is it about the lack of appreciation on the part of the patient, is it mainly about the duration of care, the fact that its end cannot be determined, the time when it has to be done, the activities that have to be performed (for example, changing your parents after defecation), or....? Social support is not only about the number of contacts, but also their intensity, quick accessibility, real understanding, or sometimes perhaps shared indignation at the ingratitude on the part of the patient. And so on and so forth. Against that background, are the instruments chosen the right ones? Do they constitute a correct operationalization of the various aspects of the basic concepts that you as researchers would like to know more about?
Another concern is the representativeness of the sample, as mentioned above. On this point, representativeness, every word is missing.
In the results section, I think the statistical results could be further translated into what actually emerges from them.
The discussion is disorderly. This is due to the fact that the introduction is not very systematically designed and the research question wording is too vague. For example, the comment that men experience a higher care burden (line 394), comes out of the blue. But if the research question had formulated the extent to which background variables (age, gender, social status, income, etc.) might influence perceived care burden, this would have been appropriate. It now seems more like an accidental finding. The same goes for the references to the literature. Very different literature is presented than in the introduction. By the way, the question applies whether comparison, for example, with the situation in Turkey, is an appropriate comparison. Can Turkish society be compared to Saudi Arabian society?
A real reflection on the limitations of the study is missing. The only one is in lines 456-457. This is too little. Reflections on the representativeness of the sample, on the appropriateness of the instruments, on the comparison of studies in other cultures with that of Saudi Arabia, on the lack of incidence and prevalence figures on informal care, and so on would be in order.
The recommendations are meager and could - if I may say so - be written down even without the study having taken place. The conclusion is particularly meager, because I would expect that this is precisely where the connection to the own situation of informal care in Saudi Arabia should be made.
Comments on the Quality of English LanguageThe English could be better, in my opinion. Some sentences are not easy to read. Some expressions are not correct (e.g. responses instead of response options).
Author Response
Please find the attachment for authors' response.

Reviewer 2 Report
Comments and Suggestions for Authors
see the attached file

Author Response

(The authors gave the same response as above.)

Reviewer 3 Report
Comments and Suggestions for Authors
I read the manuscript and found it interesting, but it was not well written. The authors need to pay attention to the comments below.
Major concerns:
T1. The title of the study is perplexing. The term "Caregiver Burden" is broad and does not indicate the specific nature of the burden (e.g., emotional, physical, financial). The phrase "in Saudi Arabia" is repeated multiple times, which is unnecessary and can be condensed. The title does not indicate what specific aspects of caregiver burden were studied (e.g., burden due to chronic illness, elderly care, etc.) make it lack specificity. The author is advised to revise or rewrite the entire title.
T2. The abstract is mostly complete but could benefit from minor adjustments to enhance clarity and completeness. The methods are described briefly, but the abstract could be improved by mentioning the specific tools used for measurement (e.g., Zarit Caregiver Burden Interview, Oslo Social Support Scale, SF-12) to give readers a clearer idea of how the data were collected. The results section is generally informative, but it could benefit from more detail on the statistical significance of the findings (e.g., p-values). Additionally, the presentation of the main results could be more focused, highlighting the most critical findings. The conclusion is clear but could be strengthened by briefly mentioning the implications of these findings, such as potential interventions or recommendations. Please do the necessary corrections on the entire abstract.
33. Introduction chapter: It is sufficiently detailed, offering a clear understanding of why caregiver burden is a significant issue globally. However, while the introduction discusses caregiver burden in a general context, it could more explicitly address the specific context of Saudi Arabia, highlighting any unique cultural or social factors that might affect caregiving in this region. To strengthen the justification for the study, the introduction should explicitly state what is not yet known or well-studied about caregiver burden in Saudi Arabia. For example, are there few studies on the relationship between caregiver burden and social support in this region? Is there a lack of understanding of how these factors interact in the context of chronic illness? Identifying these gaps will better justify the need for the current study. The rationale could be made more compelling by tying it more directly to the identified research gaps. For instance, if there is a lack of research on informal caregivers in Saudi Arabia, this should be clearly stated as a reason for conducting the study. Additionally, emphasizing the potential implications of the findings (e.g., informing policy, guiding interventions) could strengthen the rationale. The objective is well-defined, but it could be slightly reworded to reflect a more focused aim. For instance, rather than just "exploring connections," the objective could be "to determine the extent and nature of the relationship between caregiver burden, social support, and quality of life among informal caregivers in Saudi Arab.
T4. The method and tool: Generally, it looks okay. The sample calculation method appears appropriate, and the sample size is sufficiently large to provide reliable results. However, there is no mention of adjustments for potential non-response example 20% drop out rate, which is an important consideration in survey-based research. The use of convenience sampling introduces potential biases, as it may not produce a sample representative of the broader population. This limitation needs to be more explicitly addressed in the methodology, particularly in the discussion of how it might affect the generalizability of the results. The data collection process is clearly described. The use of electronic questionnaires is practical, but the methodology should also discuss how response biases were minimized (e.g., whether reminders were sent, how incomplete responses were handled). The tools used are appropriate and have been validated and shown to be reliable in previous research. This strengthens the methodological rigor of the study. However, the methodology could be enhanced by including information on whether these tools have been specifically validated in the Saudi population or similar contexts (provide detail finding on their reliability and validity results).
55. Result chapter: The results are generally well-presented, but the clarity and detail of tables, figures, and their labeling could be improved. Each result should be clearly linked to the research objectives. For example, a table title like "Table 1: Sociodemographic Characteristics of the Sample" should be specific, clear, and informative. he analysis methods used are appropriate for examining relationships between variables. However, the use of multivariate analysis (e.g., multiple regression) could provide a more comprehensive understanding of how different factors interact to influence caregiver burden, social support, and quality of life. The results generally address the study’s objectives, but the connection between the objectives and the results could be made more explicit.
66. Discussion chapter: The comparison with existing literature is adequate, but there is room for improvement. The discussion should include more updated and relevant references, particularly studies conducted in similar cultural contexts or on similar populations. The interpretation of findings is generally clear and aligns with the results presented. However, the discussion could be improved by providing a deeper analysis of the findings, including possible explanations for why certain relationships were observed (e.g., why higher caregiver burden is associated with lower quality of life and poor social support). The discussion should explore both expected and unexpected findings and consider alternative explanations where applicable. The discussion should include more updated and relevant references, particularly studies conducted in similar cultural contexts or on similar populations. The authors should ensure that each point made in the discussion directly relates to the results presented. Avoiding tangential discussions will make the narrative more coherent and focused on the study's contributions. The critical appraisal is somewhat present but could be more rigorous. The authors should critically evaluate the strengths and weaknesses of their findings, considering issues such as the potential biases introduced by the sampling method, the cross-sectional design, and the measurement tools used. The limitations are mentioned but could be more thoroughly discussed. The discussion should address how these limitations might have impacted the findings and what steps could be taken in future research to mitigate these issues.
77. Conclusion part: The summary of key findings is appropriately included in the conclusion. However, the conclusion should succinctly restate the most important results without overly repeating details already covered in the discussion. It should focus on what the findings mean in the broader context of caregiver research in Saudi Arabia.
Author Response

(The authors gave the same response as above.)

Round 2
Reviewer 1 Report
Comments and Suggestions for Authors
Dear Authors,
I was pleasantly surprised by the new version of your article. The introduction is clearer and more structured. The research questions are clear and are used as a guide throughout the article. The results are better presented; redundant figures have been omitted and tables are more clearly structured. And the discussion has been tightened up, in accordance with the research questions.
However, a few questions remain, though mostly concerning some details:
- line 245: this sentence does not flow properly
- I am missing income in Table 1; was there a reason to omit this variable?
- it seems unnecessary to me to show the percentages in tables 1 and 2 with two decimal places
- line 267: number of hours of care per day? Also adjust in table 2
- line 268 and following: these lines replace the three pie charts in the first version. It is not clear that these results do not come from tables 1 and 2.
- line 298: on what basis can socioeconomic status be listed here? Does this refer to level of education? And if so, is that really the same as socioeconomic status?
- line 363: I think male should be female!
In the discussion, I am missing some possible reflections:
- that there is only a weak correlation between caregiver burden and social support could have to do with the possibility that social support and caregiver burden have a different statistical relationship. Indeed, it could be that social support has a mediating or moderating role in the relationship between QoL and caregiver burden, or that the correlation is not linear.
- I miss a reflection on the fact that the number of male respondents is only 28%. What does that mean, in the Saudi context? And against this background, what does it mean that women in particular report a higher caregiver burden?
- The same goes for the number of single people (75%) among the respondents? What could that mean?
- and also: that the number of university graduates (also 75%) is so high among respondents. What could this mean? What does it mean that lower educated people responded much less to the questionnaire?
Not that I expect answers to these questions. But signaling the questions also raises new research questions. To get a better picture of which groups of informal caregivers face high caregiver burden and low QoL, policy can also be better crafted and adjusted. Measures can be taken that take into account specific profiles of people.
It would be good for the article if these types of questions and recommendations were mentioned, as well as suggestions for follow-up research, in which I believe qualitative research is highly desirable.
Author Response
Please see the attachment for authors' response.

Reviewer 3 Report
Comments and Suggestions for Authors
I have reviewed the revised manuscript and am completely satisfied with the work done. However, I have a few minor concerns that may require your attention. Please consider simplifying the results section to enhance reader comprehension. Additionally, could you verify the multivariate analysis presented in Table 5? It appears somewhat confusing, and I recommend consulting a statistician to assist with this. The logistic regression analysis should be presented with both univariate and multivariate findings
Author Response

(The authors gave the same response as above.)
